# Behavioral Factors Related to the Incidence of Frailty in Older Adults

**DOI:** 10.3390/jcm9103074

**Published:** 2020-09-24

**Authors:** Hiroyuki Shimada, Takehiko Doi, Kota Tsutsumimoto, Sangyoon Lee, Seongryu Bae, Hidenori Arai

**Affiliations:** 1Center for Gerontology and Social Science, National Center for Geriatrics and Gerontology, 7-430 Morioka-cho, Obu City, Aichi Prefecture 474-8511, Japan; take-d@ncgg.go.jp (T.D.); k-tsutsu@ncgg.go.jp (K.T.); sylee@ncgg.go.jp (S.L.); bae-sr@ncgg.go.jp (S.B.); 2National Center for Geriatrics and Gerontology, 7-430 Morioka-cho, Obu City, Aichi Prefecture 474-8511, Japan; harai@ncgg.go.jp

**Keywords:** frailty, older adults, cohort study, lifestyle, prevention, instrumental activities of daily living, cognitive activities, nonfrail, prefrail

## Abstract

Frailty is a widely prevalent geriatric condition whereby individuals experience age-related functional declines. This study aimed to identify behavioral factors related to the incidence of frailty in older adults. Participants were 2631 older adults (average age: 71) without physical frailty at a baseline assessment in 2011–2012 who took part in a second-wave assessment in 2015–2016. Physical frailty was defined as having limitations in at least three of the following domains: weight loss, low physical activity, exhaustion, slow walking speed, and muscle weakness. Participants completed a 16-item questionnaire examining cognitive, social, and productive activity as well as instrumental activities of daily living (IADL) as varying dimensions of lifestyle activity. During the follow-up period, 172 participants (6.5%) converted from nonfrail to frail. Logistic regression showed that the odds ratios (ORs) for conversion were significantly lower in the participants who had high IADL scores (OR: 0.78; 95% confidence interval (CI): 0.64–0.96), cognitive activity (OR: 0.74; 95% CI: 0.62–0.89), social activity (OR: 0.52; 95% CI: 0.43–0.63), and total activity (OR: 0.81; 95% CI: 0.75–0.87). There was no significant association between frailty and productive activity. Health care providers should recommend an active lifestyle to prevent frailty in older adults.

## 1. Introduction

The clinical condition of frailty is among the most prevalent and problematic expressions of population aging. Frailty is a multifaceted geriatric syndrome and an effect of age-related physiological decline reflecting multisystem functional decline and a reduced capacity to cope with stressors. Individuals experiencing frailty become more vulnerable to sudden shifts in health status triggered by even minor stressor events such as an infection [1]. As such, frailty is a complex concept involving a state of greater vulnerability to adverse health factors, including long-term care, disability, and an overall negative state of health. Frailty may also involve psychological, emotional, and social dimensions in addition to physical symptoms such as deterioration of motor performance [2].

Between a quarter and a half of people older than 85 years are estimated to be frail, and thus, have a considerably increased risk of long-term care, disability, and death [3,4]. Reducing or eliminating risk factors and increasing protective factors are potential actions for minimizing the chances of frailty. Regular physical, cognitive, social, and productive activities promote improvements in both physical and psychological health and contribute to the reversal of detrimental effects of chronic diseases as well as the maintenance of functional autonomy in older adults [5,6,7].

In 2019, individuals who were 65 years of age or older comprised 28.6% of Japan’s population [8]. Faced with an increasingly aging population, the Japanese government has been reforming its policies to provide for the needs of the elderly population and to prevent the need for long-term care. Among such efforts, the National Center for Geriatrics and Gerontology-Study of Geriatric Syndromes (NCGG-SGS) has been striving to identify behavioral factors that contribute to healthy aging. Using the NCGG-SGS database, it was confirmed that reversible factors such as lifestyle activities are associated with mild cognitive impairment (MCI) reversion in elderly individuals. A logistic regression model showed significantly higher odds ratios (ORs) for MCI reversion in participants who had an active lifestyle compared to the inactive participants [7].

This study examined the relationship between behavioral factors and the incidence of frailty in community-living older adults. The reduction in risk factors and the promotion of protective factors are essential to developing and implementing effective interventions to sustain successful aging and prevent disability. Observational studies have indicated several common factors related to disability prevention. Specifically, older persons who regularly participate in activities of daily life have shown a lower risk of disability and dementia [9,10,11,12,13,14,15,16,17]. Such findings indicate that lifestyle activities could be intermediaries in the incidence of frailty. Accordingly, the study also examined the relationship between frailty and lifestyle activities, including cognitive activity, social activity, productive activity, and instrumental activities of daily living (IADL). In this study, the hypothesis was that individuals with a loss of lifestyle activities would have a higher risk of frailty incidence than those with an active lifestyle.

## 2. Methods

### 2.1. Participants

The current study is part of a national study that assessed 5104 individuals, aged 65 years and older (average age: 71 years), enrolled in the NCGG-SGS [18]. Participants were recruited from Obu, a residential suburb of Nagoya. The inclusion criteria were residing in Obu and being 65 years or older during the baseline assessment (2011 or 2012). Some participants were excluded based on previous reports that certain conditions could produce characteristics of disability [19]. Accordingly, participants with a functional decline in basic activities of daily living (ADL) (*n* = 39), certified long-term care insurance (*n* = 126), and a history of Parkinson’s disease (*n* = 21) or Alzheimer’s disease (*n* = 8) were excluded. The study also excluded participants with missing data values of confounding factors (*n* = 323) or activity measurements (*n* = 81) and participants with frailty at baseline (*n* = 326). The study also excluded participants without follow-up measurements (*n* = 1549, reason unknown). This study analyzed data from a total of 2631 eligible older adults (mean age: 71.0 ± 4.7 years; 49.5% male) from the initial 5104 participants who took part in a follow-up assessment between August 2015 and August 2016. Table 1 shows the baseline characteristics of the participants, both included and excluded in the study. The study protocol was approved by the ethics committee of the National Center for Gerontology and Geriatrics (numbers 523 and 791), and participants gave written informed consent prior to their inclusion.

### 2.2. Operational Definition of Physical Frailty

The assessments were performed by well-trained assessors with nursing, allied health, or similar statements. Prior to commencement, all assessors acquired training from the authors in the correct protocols for administering the measurements.

The physical frailty phenotype was defined as existing limitations in three or more of the following domains: strength, mobility, physical activity, exhaustion, and weight loss. Grip strength was assessed using a handheld dynamometer (GRIP-D; Takei Ltd., Niigata, Japan). Low grip strength was determined according to a sex-specific cutoff (male: <26 kg; female: <17 kg) [20]. Walking speed as an indicator of mobility was measured using a stopwatch. The participants were asked to walk at a comfortable pace on a flat, straight surface of a 2.4 m path with a 2 m section to be traversed prior to the start marker. Low mobility was established as <1.0 m/s [21,22]. Physical activity was assessed with the following questions: (1) “Do you engage in moderate levels of physical exercise or sports aimed at health?” and (2) “Do you engage in low levels of physical exercise aimed at health?” If participants answered “no” to both, they were considered to engage in low levels of activity [21]. Exhaustion was evaluated as being present if the participant responded “yes” to the following question included on the Kihon Checklist [23], a self-reported health checklist developed by the Japanese Ministry of Health, Labour and Welfare: “In the last two weeks, have you felt tired without a reason?” Weight loss was determined as a response of “yes” to the question “Have you lost 2 kg or more in the past six months?” [23]. The participants with impairments in one or two of the five domains were considered prefrail.

### 2.3. Measurements of Lifestyle Activity

The participants completed a 16-item questionnaire examining cognitive, social, and productive activities as well as IADL as varying dimensions of lifestyle activity [7]. The following questions measured IADL: (1) “Do you go outdoors using the bus and train?”, (2) “Do you engage in cash handling and banking?”, (3) “Do you drive a car?”, and (4) “Do you use maps to go to unfamiliar places?” The following items measured cognitive activity: (5) “Do you read books or newspapers?”, (6) “Do you engage in cognitive stimulation such as board games and learning?”, (7) “Do you engage in cultural classes?”, and (8) “Do you use a personal computer?” The following questions measured social activity: (9) “Do you talk with other people every day?”, (10) “Are you sometimes called on for advice?”, (11) “Do you attend meetings in the community?”, and 12) “Do you engage in hobbies or sports activities?” Finally, the following items measured productive activity: (13) “Do you engage in housecleaning?”, (14) “Do you engage in fieldwork or gardening?”, (15) “Do you take care of grandchildren or pets?”, and (16) “Do you engage in paid work?” Answers of “yes” were considered positive responses. The total score for the 16 items (range 0–16) was calculated along with subscore totals (range 0–4) for IADL, cognitive activity, social activity, and productive activity.

### 2.4. Potential Confounding Factors

Possible confounding factors of ADL limitations were demographic variables (age, sex, and education), overweight or underweight, primary diseases or health conditions, Mini-mental State Examination (MMSE) scores [24], scores on the 15-item version of the Geriatric Depression Scale (GDS-15) [25,26], and serum albumin levels (Table 2) [27,28,29]. The overweight and underweight were determined by measuring body mass index (BMI), and the Asian cut points of overweight and underweight were set at 27.5 and <18.5 kg/m^2^, respectively [29]. Primary diseases and other health conditions—namely, stroke, heart disease, pulmonary disease, osteoarthritis hypertension, diabetes mellitus, history of falls, and medication—were acquired via self-reporting and interview surveys.

### 2.5. Statistical Analysis

The study calculated incidence rates of frailty per 1000 person-years, and compared incidence of frailty between the participants categorized as robust and prefrail at baseline using chi-square tests. Baseline characteristics were compared according to frailty status using Student’s *t*-tests and Pearson’s chi-square tests. Baseline characteristics were also compared between included and excluded participants using Student’s *t*-tests and Pearson’s chi-square tests to evaluate possible selection bias. Chi-square tests were used to compare frailty incidence between age group and sex, and adjusted standardized residuals were used to identify the impact of age on the incidence of frailty. The adjusted standardized residuals followed the *t* distribution: >1.96, *p* < 0.05 and >2.56, *p* < 0.01. 

Associations between each lifestyle activity status and incidence of frailty were analyzed with multiple logistic regression models adjusted for confounding factors (model 1). The logistic models included estimated adjusted ORs and their 95% confidence intervals (95% CIs). To determine which lifestyle activities are independently associated with frailty development, another logistic model was created including all types of activities and confounding factors (model 2). All data management and statistical computations were performed using the IBM SPSS Statistics 24.0 software package (IBM Japan, Tokyo). The significance threshold was set at 0.05.

## 3. Results

At baseline, the frailty status of the participants was 1340 (50.9%) nonfrail participants and 1291 (49.1%) prefrail participants. Among people without frailty at baseline who survived during the 4-year follow-up, 172 participants (6.5%) became frail. The incidence rate of frailty was 16.3 (95% CI: 14.1–19.0) cases per 1000 person-years. During the follow-up, 33 participants (2.5%) who were robust at baseline and 139 participants (10.8%) who were prefrail at baseline developed frailty. The frailty incidence rates among the robust and prefrail participants were 6.2 (95% CI: 4.4–8.6) and 26.9 (95% CI: 22.8–31.8) cases per 1000 person-years, respectively. There was a significant difference in the incidence rates based on the baseline status (*p* < 0.01). The comparison of the baseline characteristics of the excluded participants showed that they had higher age, lower education, higher rates of abnormal body composition, chronic diseases, history of falls, higher number of medications, lower MMSE score, higher depressive mood, lower albumin level, and lower activity status than the included participants (Table 1).

Table 2 presents potential confounding factors for frailty incidence among the participants. Significant differences based on frailty status were found for age, educational level, overweight, stroke, diabetes, fall history, medications, MMSE scores, GDS scores, serum albumin levels, and all lifestyle activities (Table 2).

Results showed that 172 participants (6.5%) had incident frailty during the 4-year follow-up period. It was found that the incidence of physical frailty increased with age (*p* < 0.01) (Figure 1A). In the residual analyses, the participants aged 65 to 69 years showed low incidence (*p* < 0.01), and the participants aged 75 to 79 years, 80 to 84 years, and 85 years and over showed a high incidence of frailty. However, there were no significant sex-specific differences in the incidence of frailty (Figure 1B). Chi-square tests identified a significantly lower incidence of prefrailty in the participants who performed all IADL activities (all activities vs. 0 to 3 activities: 42.6% vs. 55.2%, *p* < 0.001), cognitive activities (all activities vs. 0 to 3 activities: 32.6% vs. 52.3%, *p* < 0.001), and social activities (all activities vs. 0 to 3 activities: 39.4% vs. 57.2%, *p* < 0.001). However, there was no significant difference based on productive activities (all activities vs. 0 to 3 activities: 46.0% vs. 49.6%, *p* = 0.205).

The regression identified several significant relations between frailty incidence and lifestyle activities (Table 3). The individuals engaging in the following activities had lower ORs for frailty incidence in model 1: going out using the bus or train (OR: 0.47; 95% CI: 0.28–0.81), using maps to go to unfamiliar places (OR: 0.66; 95% CI: 0.46–0.95), cognitive stimulation (OR: 0.69; 95% CI: 0.48–0.98), culture lessons (OR: 0.64; 95% CI: 0.45–0.93), personal computer use (OR: 0.57; 95% CI: 0.36–0.90), daily conversation (OR: 0.48; 95% CI: 0.25–0.95), giving advice (OR: 0.52; 95% CI: 0.31–0.86), community meetings (OR: 0.49; 95% CI: 0.35–0.69), and hobbies and sports (OR: 0.32; 95% CI: 0.22–0.46). Furthermore, the multiple logistic model included all types of activities; the regression model identified several significant relationships between frailty incidence and lifestyle activities. The individuals engaging in the following activities had lower ORs for frailty incidence: community meetings (OR: 0.61; 95% CI: 0.42–0.88) and hobbies and sports (OR: 0.35; 95% CI: 0.23–0.52) (Table 3). 

Frailty incidence was also significantly associated with each total score in the following activity domains: IADL (OR: 0.78; 95% CI: 0.64–0.96), cognitive activity (OR: 0.74; 95% CI: 0.62–0.89), social activity (OR: 0.52; 95% CI: 0.43–0.63), and all activities (OR: 0.81; 95% CI: 0.75–0.87). However, there was no significant association between productive activity and frailty incidence (Table 4). The multiple logistic model included all types of activities; the regression model identified that total scores of social activities remained significant relationships with frailty incidence (OR: 0.55; 95% CI: 0.44–0.67) (Table 4).

## 4. Discussion

This study presents original findings regarding physical vulnerability with age and lifestyle activities among 2631 older adults in Obu, Japan. It was revealed that the incidence of physical frailty was associated with IADL, cognitive activity, and social activity and that individuals with high social activity scores had the lowest risk of frailty.

In this Japanese national cohort, 172 (6.5%) older adults developed frailty during the 4-year follow-up, and the incidence rate of frailty was 16.3 (95% CI: 14.1–19.0) cases per 1000 person-years. A previous meta-analysis that compared the incidence of frailty estimates based on different frailty definitions reported 43.4 (95% CI: 37.3–50.4) cases per 1000 person-years [30]. The participants of the present study showed a lower incidence rate of frailty than participants included in the meta-analysis [30]. A subanalysis in the meta-analysis identified that the incidence of frailty was higher in prefrail persons than in robust individuals (pooled incidence rates: 62.7 (95% CI: 49.2–79.8) vs. 12.0 (95% CI: 8.2–17.5) cases per 1000 person-years), respectively [30]. In this study, the frailty incidence rates among the prefrail and robust participants were 26.9 (95% CI: 22.8–31.8) and 6.2 (95% CI: 4.4–8.6) cases per 1000 person-years, respectively. Considering frailty status at baseline, the participants in this study had a lower incidence of frailty than previous studies. It was considered that participants in this study included a large number of healthy elderly people who lived independently in the community and that the survival effect impacted the difference in incidence. The principal difference between Fried’s frailty criteria and the frailty criteria of NCGG-SGS is the cutoff point for walking speed: in Fried’s criteria, it is set at 0.65 m/s (height ≤ 173 cm), whereas in NCGG-SGS, it is 1.0 m/s. Walking speed has been found to be a strong predictor of adverse events such as recurrent falls [31,32], disability [33,34,35,36,37,38,39], mortality [40,41], and hospitalization [34,35,37,42]. Previous studies have identified that the crucial point for predicting functional decline was 1.0 m/s in comfortable walking speed in community-dwelling older adults [34,35,37,38,39]. These findings also suggest that walking speed could be the most useful indicator for specifying frailty and the most reliable predictor of functional decline among older adults [43,44]. The low prevalence of physical frailty in participants of this study despite the higher cut-off point for walking speed may have been due to the participants’ better functional status than that of participants in the previous studies.

A growing body of evidence has indicated a close relationship between physical frailty and activity. A systematic review has stated that physical activity has a beneficial effect on muscle mass and strength or physical performance in healthy adults aged ≥ 60 years; however, an additional effect of dietary supplementation has been reported in only a limited number of studies [45]. A randomized controlled trial compared the effects of physical and cognitive training, nutritional supplementation, and combination treatments vs. control in reducing frailty among prefrail and frail older adults. The results showed that frailty status over a year-long period was reduced in all groups, including the control group (15%), but was significantly higher in the physical (OR: 4.05), cognitive (OR: 2.89), nutritional (OR: 2.98), and combination (OR: 5.00) intervention groups [46]. Luger and colleagues reported that frailty prevalence decreased to the same extent for participants who received volunteer-administered cognitive training and social support (−16%) and home-based physical training and nutrition programs (−17%) [47]. The study suggested that employing robust older people as volunteers could potentially foster community empowerment and contribute to a sustainable, beneficial health intervention for frail older people. Engagement in productive activities has been positively associated with older adults’ physical and psychological health and survival [48], and such activities that require complex physical and cognitive functioning may help postpone declines in physical performance, as well as induce psychosocial changes that could impact functioning in domains related to measures of frailty [49].

The multiple logistic regression analyses identified some significant relations between frailty and lifestyle activities in our cohort. Measurements of IADL, cognitive activity, and social activity were significantly associated with frailty incidence, although there was no significant relationship between productive activity and frailty. Regarding the baseline prefrailty status, chi-square tests identified a significantly lower incidence of prefrailty in the participants who performed all activities in the IADL, cognitive activity, and social activity domains, although there was no significant difference in productive activity domain. It was considered that the prevalence of prefrailty at baseline was similar for the older adults with and without all the productive activities, which have an impact on the future incidence of frailty.

A meta-analysis of longitudinal studies identified the biological, sociodemographic, physical, psychological, and lifestyle-related risk and protective factors that have been associated with frailty among older adults [50]. Significant sociodemographic factors included age, ethnicity, neighborhood, and access to private insurance; significant lifestyle factors included a better diet quality, particularly higher habitual dietary resveratrol exposure, and higher fruit/vegetable consumption; significant psychological factors included depressive symptoms; important physical factors included obesity and ADL functional status; and significant biological factors included serum uric acid [50]. The present findings revealed that, in addition to these factors identified in previous studies, the implementation of activities such as IADL and cognitive or social activities has a preventive effect on frailty incidence in older adults. Furthermore, the multiple logistic model included all types of activities; the model identified that the individuals engaging in the community meetings or hobbies and sports had a lower risk of frailty incidence than those who were not engaged in such activities. These results suggest that specific social activities may be effective to prevent frailty. Future intervention studies will need to verify whether the implementation of these activities is effective in preventing the incidence of frailty in older adults.

An important limitation of the study is that NCGG-SGS participants were not randomly recruited, which could have led to an underestimation of frailty prevalence, as the participants were relatively healthy older persons who could access health checkups from their homes. Second, the study could not perform a follow-up survey in 1549 individuals (30.3%) who enrolled in the NCGG-SGS, which could have led to an underestimation of frailty incidence due to survival effects and poor health conditions at baseline. Moreover, the comparison of the baseline characteristics of the excluded participants showed that they were older, had a lower education, had higher rates of abnormal body composition, chronic diseases, and a history of falls, used a higher number of medications, and had a lower MMSE score, higher depressive mood, lower albumin level, and lower activity status than the included participants. The nonrandom missing data could have biased the study’s findings and decreased its statistical power. A simulation study suggested that the missing not-at-random mechanism provided seriously biased estimates of the OR that moved toward zero as loss to follow-up increased. With merely 20% of the data lost to follow-up, the true odds ratio was underestimated by approximately half its value [51]. This underestimation may be the reason for no significant association between productive activity and frailty incidence in the present study. Third, participants were restricted to the Nagoya area and may not reflect national trends. Living environments were also not addressed, such as urban vs. rural or suburban and assisted living vs. living on one’s own. Finally, this study failed to address covariates related to biological factors (e.g., cytokines) and social support that could contribute to the incidence of frailty. Future studies should include these potential covariates for frailty. However, an important strength of the study is the large size of the cohort and that the findings are supported by comprehensive geriatric assessments used as indicators of frailty. In addition, to our knowledge, this is the first study to use a large population-based sample to identify the relationships between frailty incidence and lifestyle activity in an Asian population. This study suggests that modifiable lifestyle activities may be useful for preventing the incidence of frailty in older adults.

## 5. Conclusions

During a follow-up period from 2011–2012 to 2015–2016, 172 participants (6.5%) converted from nonfrail to frail. The incidence of frailty was related to lifestyle activities. The incidence of frailty was associated with IADL, cognitive activities, and social activities; therefore, it was considered that activity assessments could be used in research fields of gerontology and in primary health care settings as an indicator of disability prevention in older adults. Further intervention studies using these activities are required to determine the validity of these measures as correlates of physical frailty. Health care providers should recommend an active lifestyle, such as conducting IADL and cognitive or social activities, to prevent frailty in older adults without frailty.

## Figures and Tables

**Figure 1 jcm-09-03074-f001:**
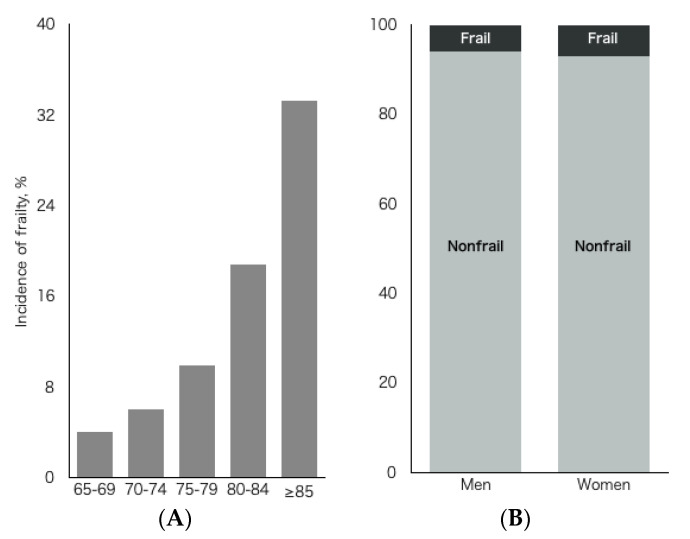
Incidence of frailty by age and sex during the 4-year follow-up period. (**A**) showed the incidence of frailty by age group. (**B**) showed sex-specific differences in the incidence of frailty.

**Table 1 jcm-09-03074-t001:** Baseline characteristics of the included and excluded participants.

	Included Participants (*n* = 2631)	Excluded Participants (*n* = 2473)	*p*
Age (years) ^†^	71.0 (4.7)	73.6 (6.4)	<0.01
Sex (% male)	49.5	48.2	0.34
Education (years) ^†^	11.7 (2.5)	10.9 (2.6) *^14^	<0.01
BMI > 27.5 (% yes)	7.6	10.7	<0.01
BMI < 18.5 (% yes)	3.3	5.8	<0.01
Stroke (% yes)	4.0	7.1	<0.01
Heart disease (% yes)	15.6	18.4	<0.01
Pulmonary disease (% yes)	10.9	12.1	0.20
Osteoarthritis (% yes)	13.8	15.4	0.11
Diabetes mellitus (% yes)	11.4	15.6	<0.01
History of falls (% yes)	13.1	17.9	<0.01
Medication (number) ^†^	1.9 (1.9)	2.4 (2.4)	<0.01
Minimental state examination (points) ^†^	26.6 (2.5)	25.6 (3.0) *^13^	<0.01
Geriatric depression scale (points) ^†^	2.4 (2.4)	3.5 (2.9) *^18^	<0.01
Serum albumin (g/dL) ^†^	4.3 (0.2)	4.3 (0.3) *^39^	<0.01
**Instrumental activities of daily living (% yes)**			
Going outdoors using bus and train	93.1	84.5	<0.01
Cash handling and banking	90.7	86.6	<0.01
Driving a car	76.7	60.2	<0.01
Using map to go unfamiliar places	67.6	53.9	<0.01
**Cognitive activity (% yes)**			
Reading of book or newspaper	97.1	93.3	<0.01
Cognitive stimulation such as board game and learning	53.7	42.2	<0.01
Culture lesson	46.8	32.6	<0.01
Using personal computer	39.6	25.2	<0.01
**Social activity (% yes)**			
Daily conversation	97.1	93.9	<0.01
Giving someone a helping hand	93.7	87.3	<0.01
Attending a meeting in the community	55.5	44.3	<0.01
Hobby or sports activity	79.9	61.5	<0.01
**Productive activity (% yes)**			
Housecleaning	87.7	84.2	<0.01
Field work or gardening	75.5	66	<0.01
Taking care of grandchild or pet	57.8	50.1	<0.01
Working	32.2	25.5	<0.01
**Activity score (points) ^†^**			
Instrumental activities of daily living	3.3 (0.8)	2.9 (1.1) *^40^	<0.01
Cognitive activity	2.4 (1.1)	1.9 (1.0) *^31^	<0.01
Social activity	3.3 (0.8)	2.9 (1.0) *^71^	<0.01
Productive activity	2.5 (0.9)	2.3 (1.0) *^38^	<0.01
**Total**	11.4 (2.4)	9.9 (2.9) *^151^	<0.01

* number of missing values; ^†^ average (standard deviation); BMI: body mass index.

**Table 2 jcm-09-03074-t002:** Comparisons of baseline characteristics according to frailty status.

	Participants without Frailty (*n* = 2459)	Participants with Frailty (*n* = 172)	*p* Value
Age (years) *	70.8 (4.5)	73.9 (5.9)	<0.01
Sex (% male)	49.8	44.8	0.20
Education (years) *	11.8 (2.5)	10.7 (2.4)	<0.01
BMI > 27.5 (% yes)	7.1	15.7	<0.01
BMI < 18.5 (% yes)	3.3	3.5	0.89
Stroke (% yes)	3.6	9.9	<0.01
Heart disease (% yes)	15.5	16.9	0.63
Pulmonary disease (% yes)	10.7	15.1	0.07
Osteoarthritis (% yes)	13.8	13.4	0.87
Diabetes mellitus (% yes)	11.0	16.9	0.02
History of falls (% yes)	12.3	25.0	<0.01
Medication (number) *	1.8 (1.9)	2.5 (2.1)	<0.01
Mini-mental state examination (points) *	26.7 (2.4)	25.3 (3.2)	<0.01
Geriatric depression scale (points) *	2.3 (2.3)	3.6 (2.7)	<0.01
Serum albumin *	4.3 (0.2)	4.3 (0.2)	<0.01
Activity score (points) *			
Instrumental activities of daily living	3.3 (0.8)	2.9 (1.0)	<0.01
Cognitive activity	2.4 (1.0)	1.9 (1.0)	<0.01
Social activity	3.3 (0.8)	2.7 (1.0)	<0.01
Productive activity	2.5 (0.9)	2.3 (0.9)	<0.01
Total	11.6 (2.3)	9.7 (2.7)	<0.01

* average (standard deviation); BMI: body mass index.

**Table 3 jcm-09-03074-t003:** Relationships between frailty status and lifestyle activities.

	Model 1		Model 2	
	Odds Ratio (95% CI)	*p* Value	Odds Ratio (95% CI)	*p* Value
**Instrumental activities of daily living (% yes)**				
Going outdoors using bus and train	0.47 (0.28–0.81)	<0.01	0.66 (0.37–1.18)	0.16
Cash handling and banking	0.72 (0.42–1.21)	0.21	0.86 (0.49–1.51)	0.60
Driving a car	1.09 (0.72–1.65)	0.68	1.33 (0.86–2.05)	0.20
Using map to go unfamiliar place	0.66 (0.46–0.95)	0.02	0.81 (0.55–1.18)	0.27
**Cognitive activity (% yes)**				
Reading of book or newspaper	0.99 (0.42–2.32)	0.98	1.40 (0.57–3.44)	0.47
Cognitive stimulation such as board game and learning	0.69 (0.48–0.98)	0.04	1.04 (0.7–1.55)	0.86
Culture lessons	0.64 (0.45–0.93)	0.02	1.12 (0.73–1.72)	0.61
Using personal computer	0.57 (0.36–0.90)	0.02	0.68 (0.42–1.09)	0.11
**Social activity (% yes)**				
Daily conversation	0.48 (0.25–0.95)	0.03	0.66 (0.32–1.37)	0.27
Giving someone advice	0.52 (0.31–0.86)	0.01	0.71 (0.41–1.22)	0.22
Attending a meeting in the community	0.49 (0.35–0.69)	<0.01	0.61 (0.42–0.88)	0.01
Hobby or sports activity	0.32 (0.22–0.46)	<0.01	0.35 (0.23–0.52)	<0.01
**Productive activity (% yes)**				
Housecleaning	0.82 (0.49–1.38)	0.45	1.04 (0.60–1.80)	0.90
Field work or gardening	0.82 (0.57–1.18)	0.53	0.99 (0.67–1.46)	0.97
Taking care of grandchild or pet	0.83 (0.60–1.16)	0.28	0.94 (0.66–1.33)	0.73
Working	0.93 (0.63–1.39)	0.74	0.88 (0.58–1.33)	0.54

Model 1 included each lifestyle activity and confounding factors which were age, sex, education, and overweight or underweight, stroke, heart disease, pulmonary disease, osteoarthritis hypertension, diabetes mellitus, history of falls, medication, Mini-mental State Examination, Geriatric Depression Scale, and serum albumin levels. Model 2 included all types of activities and confounding factors.

**Table 4 jcm-09-03074-t004:** Relationships between frailty status and lifestyle activities.

	Model 1		Model 2	
	Odds Ratio (95% CI)	*p* Value	Odds Ratio (95% CI)	*p* Value
Total score of IADL (point)	0.78 (0.64–0.96)	0.02	0.92 (0.74–1.14)	0.44
Total score of cognitive activity (point)	0.74 (0.62–0.89)	<0.01	0.92 (0.75–1.11)	0.38
Total score of social activity (point)	0.52 (0.43–0.63)	<0.01	0.55 (0.44–0.67)	<0.01
Total score of productive activity (point)	0.86 (0.72–1.04)	0.12	0.98 (0.81–1.19)	0.87
Total score of all activity (point)	0.81 (0.75–0.87)	<0.01		

Model 1 included each total score of lifestyle activity and confounding factors which were age, sex, education, and overweight or underweight, stroke, heart disease, pulmonary disease, osteoarthritis hypertension, diabetes mellitus, history of falls, medication, Mini-mental State Examination, Geriatric Depression Scale, and serum albumin levels. Model 2 included all types of activities and confounding factors. IADL: instrumental activities of daily living.

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
