# Peer review of "Behavioral Factors Related to the Incidence of Frailty in Older Adults"

_jcm, 2020, doi:10.3390/jcm9103074_

Round 1
Reviewer 1 Report
The authors addressed most of my concerns.
I have few further issues that may improve the quality of the work:
- Abstract: lines 24-25 report wrong OR estimates for social activities and all activities. Please to revise and correct them.
- Methods: I am not sure that BMI should be mentioned among the "demographic" factors.
- I would be consistent in the text in using the term "frailty" rather than "fragility".
- Methods - statistical analysis: please to revise this paragraph in accordance with the order of presentation of the study results.
- Results: the sentence "The baseline characteristics of the excluded participants were older, lower education, higher rates of abnormal body composition, chronic diseases, and history of falls, higher number of
medications, lower MMSE score, higher depressive mood, lower albumin level, and lower activity status than the included participants (Table 1)" is wrongly placed in page 9 and should undergo an English editing. - The sentence (lines 206-207): "Further, the multiple logistic model included all types of activities, the regression model identified several significant relations between frailty incidence and lifestyle activities." should undergo an English editing.
- Table 3 - lines 207-209: for a better clarity, I suggest to the Authors to split Table 3 in two tables. In the first one, I would present results at multivariable logistic regression for the single lifestyle activities, first not adjusted and then adjusted for each other. In the second table, I would show results of the multivariable logistic regression for the total scores of IADL, cognitive, social and productive activities, not adjusted and then adjusted for each other.
- Lines 289-292 should undergo an English editing.
- Lines 295-296: did the Authors refer to an underestimation of frailty prevalence or incidence?
Reviewer 2 Report
The minor issues were properly addressed.
Author Response
Reviewer #2:
Comment 1.
The minor issues were properly addressed.
Our Response 1.
Thank you for taking the time to review our manuscript.
Round 2
Reviewer 1 Report
The Authors addressed my concerns and substantially improved the quality of the manuscript.
This manuscript is a resubmission of an earlier submission. The following is a list of the peer review reports and author responses from that submission.
Round 1
Reviewer 1 Report
Thank you for the opportunity to review the study “Behavioral factors related to the incidence of frailty in older adults” (jcm-864187). In this study, the Authors evaluated the association between involvement in cognitive, social and productive activities, as well as instrumental activities of daily living (IADL), with the incidence of frailty. The results of 2631 older adults followed up over a 4-year period showed that the chance of becoming frail was significantly lower among those who had high IADL and who were involved in cognitive and social activities. No significant results were found for productive activity.
The large sample size and the prospective design are strengths of the study. However, I think that the study could be improved by clarifying some points of the text and revising the analyses to further support the observed findings. My main concerns are reported below:
- Introduction: I would summarize this section, in particular in page 2, lines 50-54, since I would not go in so much details about that previous study on MCI reversion.
- Methods:
- page 2, lines 75-77: please to specify which variables had missing values and, if possible, how many participants had not follow-up measurements because deceased or because lost to follow-up. To evaluate possible selection bias, I would also briefly compare the baseline characteristics of those excluded because missing data and because of no follow-up measurements, with those included in the study.
- page 4, lines 91-92: please to cite here the Fried at al (2001) manuscript on frailty phenotype. Lines 105-106: these lines repeat information stated in lines 91-92. Instead, no definition of pre-frailty is given.
- page 5, lines 125: please to specify how body mass index was measured (were body height and weight self-reported or measured?) and why a cutoff of 27.5 was used to categorize that variable.
- statistical analyses: line 135, please to revise the “prevalence rates of frailty”. Was that instead an incidence rate?
- Results:
- page 6, lines 163-171: in this section the Authors talk often about “prevalence” of frailty, but it seems that they refer to the “incidence”. Please to revise it.
- Table 3: Please to specify the confounders of that analysis in the table footnotes. Moreover, in Table 3 as well as in the analyses reported in lines 184-188, it is not clear whether each activity was tested singularly in the model. If the goal was to evaluate which activities are independently associated with frailty development, I think that the Authors should perform a further model including all types of activities.
- Discussion:
- Maybe I missed it, but it is not very clear how the Authors explain the lack of significant association between productive activities and incident frailty.
- Please to revise Discussion based on the comments above.
Author Response
Reviewers' comments:
Reviewer #1:
Comment 1.
Introduction: I would summarize this section, in particular in page 2, lines 50-54, since I would not go in so much details about that previous study on MCI reversion.
Our Response 1.
We have summarized the points you pointed out.
Location of edits 1.
Introduction
A logistic regression model showed significantly higher odds ratios (ORs) for MCI reversion in participants who had active lifestyle compared to inactive participants [7].
Comment 2.
Methods:
page 2, lines 75-77: please to specify which variables had missing values and, if possible, how many participants had not follow-up measurements because deceased or because lost to follow-up. To evaluate possible selection bias, I would also briefly compare the baseline characteristics of those excluded because missing data and because of no follow-up measurements, with those included in the study.
Our Response 2.
Unfortunately, it was not clear whether the participants without follow-up measurements decreased their function because we could not receive any response. We added that it is unclear why we were unable to follow-up the participants. We performed the analysis to compare the baseline characteristics between the included and excluded participants. Table 1 mentioned the baseline characteristics of the participants. The excluded participants showed significant differences for almost all characteristics.
Location of edits 2.
2.5. Statistical analysis
To evaluate possible selection bias, we compare the baseline characteristics between included and excluded participants using Student’s t-tests and Pearson’s chi-square tests.
4. Discussion
Second, we could not perform a follow-up survey in 1,549 individuals who enrolled in the NCGG-SGS, which could have led to an underestimation of frailty incidence due to survival effects and poor health conditions at baseline.
Table 1. Baseline characteristics of the included and excluded participants.
  |
Participants (n = 2,631) |
Excluded participants (n = 2,473) |
P |
Age (years) † |
71.0 (4.7) |
73.6 (6.4) |
<0.01 |
Sex (% male) |
49.5 |
48.2 |
0.34 |
Education (years) † |
11.7 (2.5) |
10.9 (2.6)*14 |
<0.01 |
BMI>27.5 (% yes) |
7.6 |
10.7 |
<0.01 |
BMI<18.5 (% yes) |
3.3 |
5.8 |
<0.01 |
Stroke (% yes) |
4.0 |
7.1 |
<0.01 |
Heart disease (% yes) |
15.6 |
18.4 |
<0.01 |
Pulmonary disease (% yes) |
10.9 |
12.1 |
0.20 |
Osteoarthritis (% yes) |
13.8 |
15.4 |
0.11 |
Diabetes mellitus (% yes) |
11.4 |
15.6 |
<0.01 |
History of falls (% yes) |
13.1 |
17.9 |
<0.01 |
Medication (number) † |
1.9 (1.9) |
2.4 (2.4) |
<0.01 |
Mini-mental state examination (points) † |
26.6 (2.5) |
25.6 (3.0)*13 |
<0.01 |
Geriatric depression scale (points) † |
2.4 (2.4) |
3.5 (2.9)*18 |
<0.01 |
Serum albumin (g/dL) † |
4.3 (0.2) |
4.3 (0.3)*39 |
<0.01 |
Instrumental activities of daily living (% yes) |
|||
Going outdoors using bus and train |
93.1 |
84.5 |
<0.01 |
Cash handling and banking |
90.7 |
86.6 |
<0.01 |
Driving a car |
76.7 |
60.2 |
<0.01 |
Using map to go unfamiliar place |
67.6 |
53.9 |
<0.01 |
Cognitive activity (% yes) |
|||
Reading of book or newspaper |
97.1 |
93.3 |
<0.01 |
Cognitive stimulation such as board game and learning |
53.7 |
42.2 |
<0.01 |
Culture lesson |
46.8 |
32.6 |
<0.01 |
Using personal computer |
39.6 |
25.2 |
<0.01 |
Social activity (% yes) |
|||
Daily conversation |
97.1 |
93.9 |
<0.01 |
Giving someone a helping hand |
93.7 |
87.3 |
<0.01 |
Attending a meeting in the community |
55.5 |
44.3 |
<0.01 |
Hobby or sports activity |
79.9 |
61.5 |
<0.01 |
Productive activity (% yes) |
|||
Housecleaning |
87.7 |
84.2 |
<0.01 |
Field work or gardening |
75.5 |
66 |
<0.01 |
Taking care of grandchild or pet |
57.8 |
50.1 |
<0.01 |
Working |
32.2 |
25.5 |
<0.01 |
Activity score (points) † |
|||
Instrumental activities of daily living |
3.3 (0.8) |
2.9 (1.1)*40 |
<0.01 |
Cognitive activity |
2.4 (1.1) |
1.9 (1.0)*31 |
<0.01 |
Social activity |
3.3 (0.8) |
2.9 (1.0)*71 |
<0.01 |
Productive activity |
2.5 (0.9) |
2.3 (1.0)*38 |
<0.01 |
Total |
11.4 (2.4) |
9.9 (2.9)*151 |
<0.01 |
*number of missing value, †average (standard deviation), BMI: body mass index
Comment 3.
page 4, lines 91-92: please to cite here the Fried at al (2001) manuscript on frailty phenotype. Lines 105-106: these lines repeat information stated in lines 91-92. Instead, no definition of pre-frailty is given.
Our Response 3.
Thank you for pointing out. We have revisited the part of the text that you pointed out.
Location of edits 3.
2.2. Operational definition of physical frailty
The participants with impairments in one or two of the five domains were considered prefrail.
Comment 4.
page 5, lines 125: please to specify how body mass index was measured (were body height and weight self-reported or measured?) and why a cutoff of 27.5 was used to categorize that variable.
Our Response 4.
We added the text in more detail about body composition.
Location of edits 4.
2.4. Potential confounding factors
The demographic variables “overweight” and “underweight” were determined by measuring body mass index (BMI), and the Asian cut points of overweight and underweight were set at 27.5 kg/m2 and < 18.5 kg/m2, respectively [29].
Comment 5.
statistical analyses: line 135, please to revise the “prevalence rates of frailty”. Was that instead an incidence rate?
Our Response 5.
As you pointed out. We have corrected the text.
Location of edits 5.
We calculated incidence rates of frailty per 1,000 person-years, and we compared incidence of frailty between the participants categorized as robust and prefrail at baseline and age- and sex-specific incidence rates of frailty using chi-square tests.
Comment 6.
- Results:
page 6, lines 163-171: in this section the Authors talk often about “prevalence” of frailty, but it seems that they refer to the “incidence”. Please to revise it.
Our Response 6.
We revised the term of the section and related sentences prevalence to incidence.
Location of edits 6.
Results showed that 172 participants (6.5%) had incident frailty during the 4-year follow-up period. It was found that the incidence of physical frailty increased with age (p < .01) (Figure 1, left). In the residual analyses, the participants aged 65 to 69 years showed low incidence (p < .01), and the participants aged 75 to 79 years, 80 to 84 years, and 85 years and over showed high incidence of frailty. However, there were no significant sex-specific differences in the incidence of frailty (Figure 1, right). Chi-square tests identified a significantly lower incidence of prefrailty in the participants who performed all IADL activities (all activities vs. 0 to 3 activities: 42.6% vs. 55.2%, p < 0.001), cognitive activities (all activities vs. 0 to 3 activities: 32.6% vs. 52.3%, p < 0.001), and social activities (all activities vs. 0 to 3 activities: 39.4% vs. 57.2%, p < 0.001). However, there was no significant difference based on productive activities (all activities vs. 0 to 3 activities: 46.0% vs. 49.6%, p = 0.205).
Figure 1. Incidence of frailty by age and sex during the 4-year follow-up period
4. Discussion
Regarding the baseline prefrailty status, chi-square tests identified a significantly lower incidence of prefrailty in the participants who performed all activities in the IADL, cognitive activity, and social activity domains, although there was no significant difference in productive activity domain.
Comment 7.
Table 3: Please to specify the confounders of that analysis in the table footnotes. Moreover, in Table 3 as well as in the analyses reported in lines 184-188, it is not clear whether each activity was tested singularly in the model. If the goal was to evaluate which activities are independently associated with frailty development, I think that the Authors should perform a further model including all types of activities.
Our Response 7.
We mentioned the confounders in the table footnotes ant that each activity was tested singularly in the model. We performed a further logistic model including all types of activities and confounding factors.
Location of edits 7.
2.5. Statistical analysis
Associations between each lifestyle activity status and incidence of frailty were analyzed with logistic regression models, and confounding factors were addressed with a multiple adjustment model including estimated adjusted ORs and their 95% confidence intervals (95% CIs). To determine which lifestyle activities are independently associated with frailty development, further logistic model was performed including all types of activities and confounding factors.
3. Results
Further, the multiple logistic model included all types of activities, the regression model identified several significant relations between frailty incidence and lifestyle activities. The individuals engaging in the following activities had lower ORs for frailty incidence: community meetings (OR: 0.61, 95% CI: 0.42–0.88) and hobbies and sports (OR 0.35, 95% CI 0.23–0.52).
Footnote of Table 3.
This model included each lifestyle activity or total score of lifestyle activities and confounding factors which were age, sex, education, and overweight or underweight, stroke, heart disease, pulmonary disease, osteoarthritis hypertension, diabetes mellitus, history of falls, medication, Mini-Mental State Examination, Geriatric Depression Scale, and serum albumin levels. IADL:activities of daily living,
4. Discussion
Further, the multiple logistic model included all types of activities, the model identified the individuals engaging in the community meetings or hobbies and sports had lower risk for frailty incidence than those who were not engaged in such activities. These results suggest that specific social activities may be effective to prevent frailty.
Comment 8.
- Discussion:
Maybe I missed it, but it is not very clear how the Authors explain the lack of significant association between productive activities and incident frailty. Please to revise Discussion based on the comments above.
Our Response 8.
We mentioned explain the lack of significant association between productive activities and incident frailty in the discussion section.
Location of edits 8.
4. Discussion
Regarding the baseline prefrailty status, chi-square tests identified a significantly lower incidence of prefrailty in the participants who performed all activities in the IADL, cognitive activity, and social activity domains, although there was no significant difference in productive activity domain. It was considered that the prevalence of prefrailty at baseline was similar for the older adults with and without all the productive activities, which have an impact on the future incidence of frailty.
Reviewer 2 Report
Well conducted and precise article considering the Behavioral factors related to the incidence of frailty in older adults.
I would consider more specificity in the following lines:
line 32 - what kind of stressors?
line 36 - what kind of symptoms?
Author Response
Reviewer #2:
Comment 1.
Well conducted and precise article considering the Behavioral factors related to the incidence of frailty in older adults. I would consider more specificity in the following lines:
line 32 - what kind of stressors?
Our Response 1.
Vulnerability of frail older people to a sudden change in health status following a minor illness. The green line represents a fit older person who, following a minor stress such as an infection, experiences a relatively small deterioration in function and then returns to homeostasis. The red line represents a frail older person who, following a similar stress, experiences a larger deterioration which may manifest as functional dependency and who does not return to baseline homeostasis.
Location of edits 1.
Individuals experiencing frailty become more vulnerable to sudden shifts in health status triggered by even minor stressor events such as an infection.
Comment 2.
line 36 - what kind of symptoms?
Our Response
We explained briefly physical symptoms.
Location of edits
Frailty may also involve psychological, emotional, and social dimensions in addition to physical symptoms such as deterioration of motor performance.